https://doi.org/10.1038/s41467-021-21125-3　　**OPEN**

# Winner-takes-all resource competition redirects cascading cell fate transitions

Rong Zhang[1], Hanah Goetz [1], Juan Melendez-Alvarez [1], Jiao Li[1,2], Tian Ding[2], Xiao Wang [1] & Xiao-Jun Tian [1]✉

Failure of modularity remains a significant challenge for assembling synthetic gene circuits with tested modules as they often do not function as expected. Competition over shared limited gene expression resources is a crucial underlying reason. It was reported that resource competition makes two seemingly separate genes connect in a graded linear manner. Here we unveil nonlinear resource competition within synthetic gene circuits. We first build a synthetic cascading bistable switches (Syn-CBS) circuit in a single strain with two coupled self-activation modules to achieve two successive cell fate transitions. Interestingly, we find that the in vivo transition path was redirected as the activation of one switch always prevails against the other, contrary to the theoretically expected coactivation. This qualitatively different type of resource competition between the two modules follows a 'winner-takes-all' rule, where the winner is determined by the relative connection strength between the modules. To decouple the resource competition, we construct a two-strain circuit, which achieves successive activation and stable coactivation of the two switches. These results illustrate that a highly nonlinear hidden interaction between the circuit modules due to resource competition may cause counterintuitive consequences on circuit functions, which can be controlled with a division of labor strategy.

---

[1] School of Biological and Health Systems Engineering, Arizona State University, Tempe, AZ, USA. [2] Department of Food Science and Nutrition, Zhejiang University, Hangzhou, Zhejiang, China. ✉email: xiaojun.tian@asu.edu

Modularity is an important design principle for engineering sophisticated synthetic gene circuits by breaking the system down into small modules to reduce complexity. However, the whole circuit often does not function as expected when the tested modules are assembled, even after several rounds of design-build-test iterations. One of the most important reasons for a high rate of device failure is that various hidden circuit-host interactions, including resource competition, could significantly perturb the performance of synthetic gene circuits[1–4]. The available cellular resources in the host cell, such as transcriptional and translational machinery (e.g., RNA polymerases and ribosomes), are limited for synthetic gene circuits[5,6], thus resulting in undesired competition between the modules within one gene circuit. For example, resource competition causes retroactivity from downstream regulators to upstream dynamics and alters the expected behaviors[7,8]. Thus, it is essential to characterize how the modules in one circuit are unintentionally coupled because of the limited amount of shared resources and how this coupling leads to modularity loss.

The coupling between two separated genes in the same plasmid is found to be constrained by an inverse linear relationship, analogous to the isocost lines used in economics[9,10]. The dependence between genetic loads and gene expression is also found to be governed by equations analogous to Ohm's law used in electrical circuits[11]. These elegant equations indeed help us to quantitatively understand resource competition between simple circuits. However, the behavior between more complex modules within one gene circuit, such as positive feedback loops, remains unclear.

In this work, we aim to build synthetic cascading bistable switches (Syn-CBS) circuits to achieve successive cell fate transitions. First, we construct a single strain Syn-CBS circuit with two mutually connected self-activation modules to achieve stepwise activation of two bistable switches by controlling the inducer dose. Interestingly, we find that an increase of inducer changes the system from a state in which only one of the switches is activated to a state in which the other is the only one activated, instead of the theoretically expected coactivation of both switches. The underlying reason is that the two modules compete for limited resources and thus inhibit each other indirectly. This "winner-takes-all (WTA)" behavior resulting from resource competition between the two connected modules is verified in controlled experiments when the modules were disconnected. We further find the relative strength of connections between the modules determines the winner. To decouple the resource competition between the two modules, we construct a two-strain Syn-CBS circuit that could achieve stable coactivation of the two coupled bistable switches. Thus, the effect of the resource competition on the Syn-CBS circuit is minimized through a division of labor using microbial consortia.

## Results

**Design of Syn-CBS circuit to achieve successive cell fate transitions.** The existence of multiple stable states under the same condition, also known as multistability, plays a critical role in diverse biological processes[12–20]. Previously, we mathematically predicted and experimentally verified that epithelial-to-mesenchymal transition is a two-step process governed by cascading bistable switches (CBS)[16,17]. To further understand the design principle of CBS for achieving successive cell fate transitions, we designed a Syn-CBS circuit (circuit CT61) with two mutually regulated modules. In this design (Fig. 1a), one module (M1) is designed with self-activation of AraC, which is controlled by Arabinose (L-ara). The other module (M2) is designed with self-activation of LuxR and is controlled by quorum-sensing

signal 3oxo-C6-HSL (C6). GFP with LVA tag (GFP-lva) and RFP with AAV tag (RFP-aav) are used as the outputs of the two switches, respectively. Both tags were chosen to ensure that maintenance of stable steady states was not due to GFP or RFP slow degradation. Since we had tested that both individual modules could function as bistable switches[3], we expected that with the correct connections between the two modules, we would see the successive activation of these two bistable switches. That is, with an increase in the dose of the inducer L-ara, the Syn-CBS strain should transition from a state with no activation of either switch to a state with only one switch activated, and then to a final state with both switches activated.

To demonstrate that this circuit design could achieve successive cell fate transitions, we developed a mathematical model for the Syn-CBS circuit (see Supplementary information for details). Through graphical analysis of the nullclines, vector field, and potential landscape in the M1–M2 phase plane, the model predicted that this system could achieve a stepwise activation of the two switches in two ways (Fig. 1b, c and Supplementary Fig. 1). The nullclines analysis shows that both M1 and M2 could function as a bistable switch with the other as an input (Fig. 1b, c). That is, both modules need the other to be above a certain threshold for activation (SN1 for M1 activation and SN2 for M2 activation). However, the activation thresholds depend on the strengths of the links between the two modules. If the strength of the M2-to-M1 connection is strong and the M1-to-M2 connection is weak, the dose of L-ara required for activation of the M1 switch is smaller than the dose needed for activation of the M2 switch. That is, the threshold SN1 is smaller than SN2 (Fig. 1b). The corresponding nullcline intersections give three stable steady states: LL (low-RFP/low-GFP, black circle), LH (low-RFP/high-GFP, green circle), and HH (high-RFP/high-GFP, yellow circle). With increase of the L-ara dose, both the levels of M1 and M2 increase. However, the M1 switch is activated first, turning cells green because of the low activation threshold. The M2 module is activated later, turning cells yellow (representing the coactivation of both RFP and GFP) under a larger L-ara dose. The quasi-potential landscape was calculated by solving the corresponding chemical master equation (CME) (see Supplementary information for details) to visualize the three cell fates as potential wells (Supplementary Fig. 1a, dark blue). A design with a weak M2-to-M1 connection and a strong M1-to-M2 connection leads to a flipped scenario, in which cells transition from the LL state to a HL state (high-RFP/low-GFP, red circle) and then to the HH state (Fig. 1c and Supplementary Fig. 1b). When the relative strength of the connections between the two modules is similar, we could have either a system with only the two LL and HH states, or a system with all the four states, both of which are not good for successive cell fate transitions. Thus, the Syn-CBS circuit is a good theoretical design to achieve successive cell fate transitions.

**Resource competition deviates cell fate transition from the desired stepwise manner.** Next, we constructed and put the whole Syn-CBS circuit (circuit CT61) on a medium-copy (20–30 copies) backbone into an *E. coli* strain. We first studied the relationship between the two modules by measuring the mean GFP and RFP levels at increasing arabinose concentrations analogous to the phase plane analysis using a plate reader. We found that RFP vs. GFP showed a negative relationship, as increase of one module simultaneously decreases the other (Fig. 2a). These results are opposite from the theoretical analysis that both modules positively activate each other. This suggests that there is significant resource competition between the two modules. Interestingly, this inverse relationship followed a two-phase

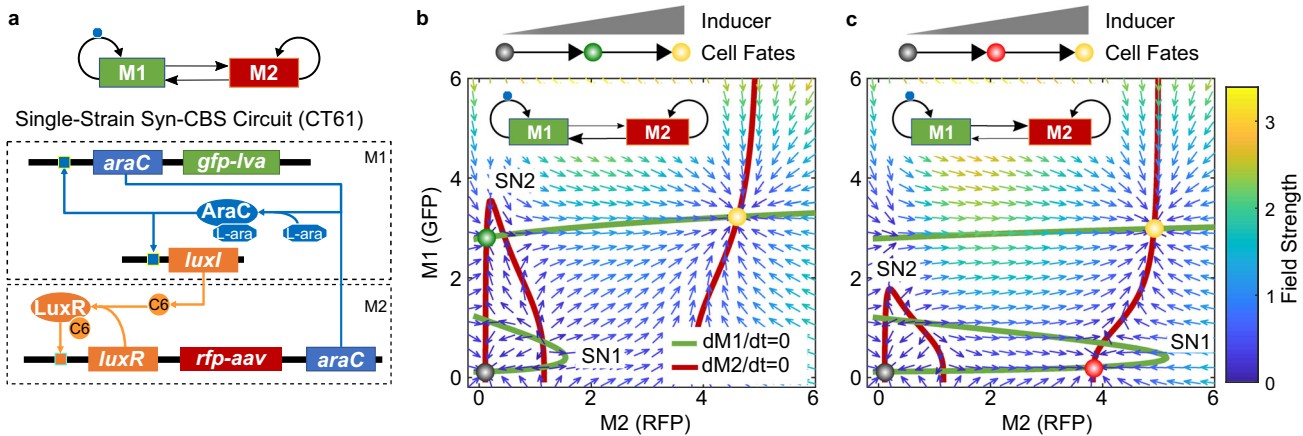

**Fig. 1 Conceptual design of the synthetic cascading bistable switches (Syn-CBS) circuit. a** Diagram of the Syn-CBS circuit, in which two self-activation modules mutually activate each other. The araC self-activation in Module 1 (M1), regulated by L-ara, is designed to achieve one bistable switch. The luxR self-activation in Module 2 (M2), regulated by C6, is designed to achieve another bistable switch. **b**, **c** Phase plane analysis shows the two different expected cell fate transition paths depending on the strength of the links between the two switches. **b** A weak M1-to-M2 link and a strong M2-to-M1 link lead to a cell fate transition from a RFP-low/GFP-low state (black circle), to a RFP-low/GFP-high state (green circle), and then to a RFP-high/GFP-high state (yellow circle). **c** A strong M1-to-M2 link and a weak M2-to-M1 link lead to a cell fate transition from a RFP-low/GFP-low state (black circle), to a RFP-high/GFP-low state (red circle), and then to a RFP-high/GFP-high state (yellow circle). The nullclines of M1 and M2 are shown in green and red, respectively. The vector field of the system is represented by small arrows, where the color is proportional to the field strength. The three cell fates are indicated by filled circles at the intersections of the two nullclines.

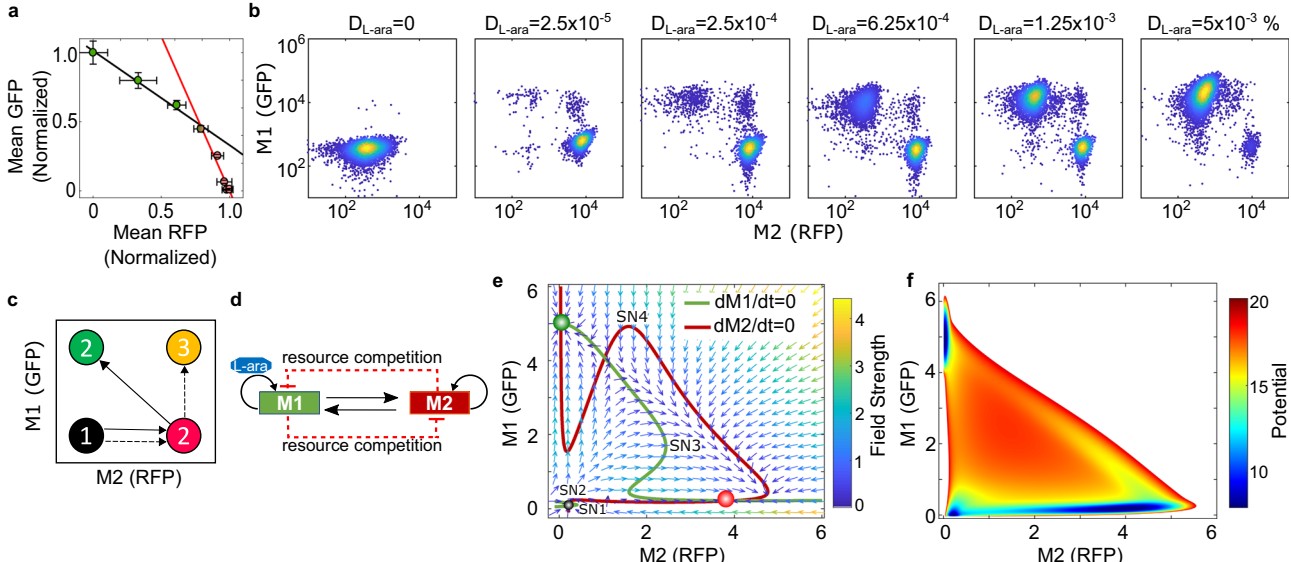

**Fig. 2 Resource competition deviates the cell fate transitions in the one-strain Syn-CBS circuit. a** The normalized steady-state signal intensity of average RFP vs. GFP measured by a plate reader shows a two-phase piecewise linear relationship. Data displayed as mean ± SD ($n = 3$ biological independent samples). **b** Flow cytometry data show cell state transitions in one-strain Syn-CBS circuit with increasing level of inducer L-ara ($D_{L\text{-}ara}$). In total, 10,000 events were recorded for each sample. Data shown from one representative of four independent biological replicates. **c** Diagram of the perturbed state transitions by resource competition. Dash line: expected path. Solid line: perturbed path. **d** Diagram of the revised model by including resource competition. **e** Phase plane diagrams. The nullclines of M1 and M2 are shown in green and red, respectively. The vector field of the system is represented by small arrows, where the color is proportional to the field strength. The three cell fates are indicated by filled circles (black, red, and green) at the intersections of the two nullclines. **f** Calculated potential landscape. Circuit CT61 was used here. Source data are provided as a Source Data file.

piecewise linear function, showing a shallow slope when the GFP level was high (black curve, Fig. 2a) and a steep slope when the RFP level was high (red curve, Fig. 2a). To further study this phenomenon, we measured cell fate transitions at the single-cell level with flow cytometry under various concentrations of L-ara. Unexpectedly, three different stable steady states were found in the RFP/GFP space (Fig. 2b). With increase of the inducer L-ara, most of the cells first entered a high-RFP/low-GFP state, then, to our surprise, jumped to a low-RFP/high-GFP state with only a

few cells staying in the high-RFP/high-GFP state (Fig. 2b and Supplementary Fig. 2). Thus, experimental data showed a negligible chance for the existence of the theoretically expected coactivation state (high-RFP/high-GFP), which we no longer consider to be a steady state. That is, the desired path of cell fate transitions was redirected from the HH state to the LH state (Fig. 2c). We also tested this Syn-CBS circuit (circuit CT61) with a low-copy backbone to study whether coactivation could be more accessible. We found that Module 1 was not activated even with a high dose

of inducer (Supplementary Fig. 3). This is most likely due to resource depletion by Module 2.

The discrepancy between the model prediction and experimental data was resolved by including resource competition into the model (see Supplementary information for details). The two modules competed for limited resources, thus indirectly inhibiting each other (red links, Fig. 2d), an idea not included in the original Syn-CBS gene circuit design. The shapes of the new nullclines change in comparison to the Syn-CBS model without resource competition (Figs. 1b and 2e, respectively). The behavior is similar in the sense that M1 activation needs M2 to be above a certain threshold (SN1 on the green curve, Fig. 2e). Nonetheless, the continuous increase of M2 now turns the M1 switch off after reaching yet another threshold (SN3 on the green curve, Fig. 2e). Additionally, the M2 nullcline now reflects that M2 can switch off as M1 increases (SN4 of the red curve, Fig. 2e). Thus, the intersections of the two nullclines give three different steady states: the LL state (black circle), LH state (green circle), and HL state (red circle). The quasi-potential landscape also shows three potential wells corresponding to three cell fates without coactivation (Fig. 2f, dark blue). Taken together, our results suggest that resource competition between the two modules in the Syn-CBS circuit (CT61) deviates cell fate transitions from the desired stepwise manner.

**WTA behavior is found in the resource competition between separated bistable switches.** To further confirm the resource competition between the two modules in the Syn-CBS circuit, we studied the behaviors of the two separated bistable switches (Syn-SBS) system (circuit IC15), in which the previous links between the two modules of the Syn-CBS circuit (circuit CT61) were removed (Fig. 3a). The network topology (Fig. 3a, right) is similar to the synthetic circuit MINPA[15] and cell differentiation system[21–24], with the difference being a mutual inhibition mediated by resource competition. The cell fate transition was induced by increasing doses of C6 combined with a fixed dose of L-ara added at the beginning of the experiment and measured with flow cytometry (Fig. 3b). It is noted that the activation speed of the bistable switch elevates while the dose of its inducer increases[3,25]. Thus, under a low dose of C6, M1 is activated so quickly that M2 is repressed from activation. Under a moderate dose of C6, the activation speeds of the two modules are similar, thus leading to their coactivation. However, a high dose of C6 activated the M2 switch so quickly that M1 was completely blocked by M2 from activation (Fig. 3b and Supplementary Fig. 4). Thus, the presence of resource competition between the two modules results in a "WTA" behavior, where the first activated switch always suppresses the activation of the second switch.

In order to understand the mechanisms of the WTA phenomena, we conducted simulation with a mathematical model for the Syn-SBS system (see Supplementary information for more details). As shown in Fig. 3c, the simulated cell fates can be represented by the levels of M1 and M2 in the dose space of two inducers. The space is divided into four regions: no switch activation in the corner with low C6 and low L-ara, M2 switch activation only in the corner with high C6 and low L-ara, M1 switch activation only in the corner with low C6 and high L-ara, and coactivation of the two switches in the corner where the two inducers are high and well-balanced. Three stable steady states, including two states with only one winner and the coactivation state can be found at the intersections of the nullclines and direction field (green, red, and yellow circles, Fig. 3d) and in the dark blue areas of the potential landscape (Fig. 3e) using inducer doses at $9.5 \times 10^{-4}$% for L-ara and $5 \times 10^{-8}$ M for C6. However, only two single-cell representative trajectories simulated from the

stochastic model (red and yellow highlighted trajectories Fig. 3d, see Supplementary information for more details) are found to be in either the coactivation or M2 activation state. This is consistent with the flow cytometry data (Fig. 3b, rightmost panel). The existence of the coactivation state in the Syn-SBS system (circuit IC15) may result from a smaller burden than that of the Syn-CBS system (circuit CT61), given that two more genes are included in the latter. It is noted that the cell becomes committed to the RFP-high state after its trajectory crosses Separatrix II (rightmost pink curve in Fig. 3d). If the levels of M1 and M2 are well-balanced, the cell is able to reach the coactivation state between the two separatrices.

To fully comprehend how sequential activation of the two bistable switches in the one-strain Syn-CBS circuit (circuit CT61) failed due to resource competition, we then studied how sequential addition of the two inducers affects the cell fate transitions in the Syn-SBS system (circuit IC15). While fixing the doses of both inducers, we added L-ara at time point 0 h but varied the addition time point of C6 from 0 to 4 h to mimic the design of the Syn-CBS circuit (Supplementary Fig. 5a). As shown in Supplementary Fig. 5b, when both inducers were added at time point 0, the M2 switch won the competition, seen by the fact that most of the cells are in the RFP-high state, consistent with Fig. 3b. However, when C6 was added 1–2 h later, more and more cells showed coactivation of both switches (Supplementary Fig. 5b, c). When C6 was added 4 h later, most of the cells showed only high-GFP (Supplementary Fig. 5b, c). That is, the M1 switch was activated first and started to repress the activation of the M2 switch by taking all the available resources. It is noted that the inactivation of the M2 switch was not due to the late addition of C6 since the M2 switch was able to be activated in the parallel experiment with the same C6 dose and time points but without L-ara (Supplementary Fig. 5b, bottom).

In addition, the simulated cell fates with a fixed L-ara dose are shown in the space of the C6 dose and the time of C6 addition, $D_{C6}$ and $T_{C6}$ (Supplementary Fig. 5d): M1 switch activation only (in the region with low C6 levels or late C6 addition), M2 switch activation only (in the region with high C6 levels), and coactivation of the two switches (in the region with moderate C6 levels and early addition). It is noted that adding a delay to C6 addition could change the cell fate from M2 activation to coactivation or even M1 activation, consistent with experimental data. Three representative stochastic single-cell trajectories with three C6 addition time points were shown in the M1–M2 phase planes (Supplementary Fig. 5e). The trajectories of the cells were first following the direction field in the M1–M2 phase plane to the GFP-high state before C6 addition (left panel), and then following the direction field in the phase plane to three different states after C6 addition (right panel). At the time point after the cell has crossed Separatrix I (uppermost pink curve, right panel of Supplementary Fig. 5e), adding C6 does not activate the M2 switch (green highlighted trajectory). At the time point where the cell has not yet crossed Separatrix I or II, adding C6 may lead to coactivation (yellow highlighted trajectory). Adding C6 early on may lead to the cell crossing Separatrix II (rightmost pink curve, right panel of Supplementary Fig. 5e) and M2 activation only (red highlighted trajectory). Taken together, these results confirm the WTA behavior when resource competition is present between the two modules.

**Relative strength of module connections determines the winner of resource competition.** To further understand how the winner is determined due to resource competition between the modules within the Syn-CBS circuit, we studied if the strength of the module connections affected the outcomes of the cell fate

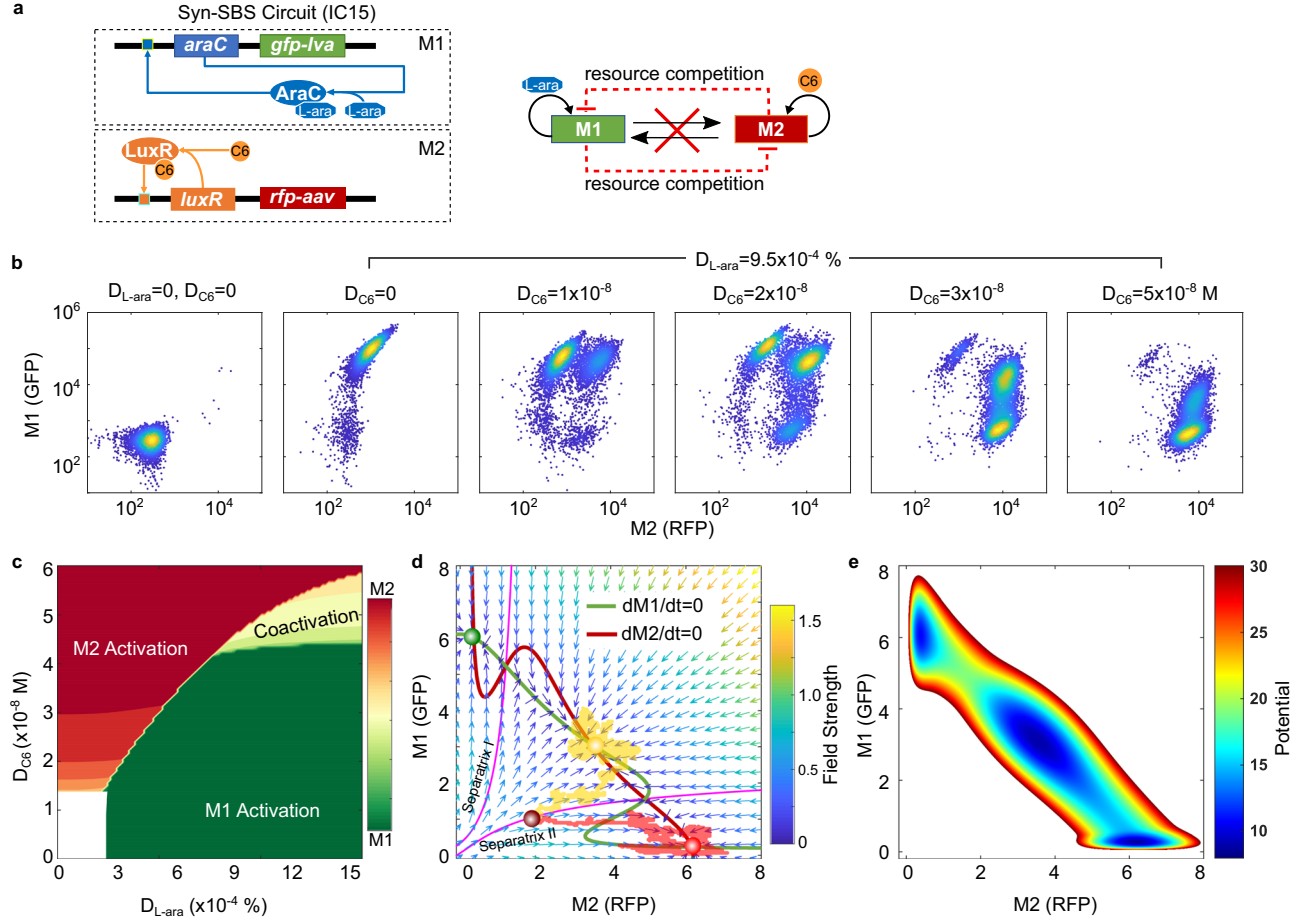

**Fig. 3 Resource competition between two separate bistable switches. a** Diagram of the two separate bistable switches (Syn-SBS). **b** Flow cytometry data show cell state transitions with an increasing level of inducer C6 ($D_{C6}$) and a fixed dose of L-ara ($D_{L-ara} = 9.5 \times 10^{-4}\%$). In total, 10,000 events were recorded for each sample. The two inducers were both added at 0 h. Data from one representative of three independent biological replicates. **c** Cell fates in the space of two inducers L-ara and C6 at addition time 0 h. **d** Simulated stochastic trajectories highlighted on the phase plane diagram. The nullclines of M1 and M2 are shown in green and red, respectively, while separatrices are shown in pink. The vector field of the system is represented by small arrows, where the color is proportional to the field strength. The three cell fates (red, green, and yellow circles) are found at the intersections of the two nullclines. Two representative single-cell stochastic trajectories (yellow and red highlights) show the evolution of the system from the same initial condition (purple circle, $D_{L-ara} = 0\%$, and $D_{C6} = 0$ M) to two different states with the same induction ($D_{L-ara} = 9.5 \times 10^{-4}\%$ and $D_{C6} = 5 \times 10^{-8}$ M). **e** Calculated potential landscape. Circuit IC15 was used here. Source data are provided as a Source Data file.

transitions by fine-tuning the M1-to-M2 link experimentally. A hybrid promoter Para/tet was used to control the production of C6 in order to tune the module connection by external chemical inducer anhydrotetracycline hydrochloride (aTc). As shown in Fig. 4a, luxI gene expression is now jointly regulated by AraC and TetR, while TetR is negatively controlled by aTc.

With the design for this hybrid Syn-CBS circuit (circuit CT81), we fixed the L-ara dose to $9.5 \times 10^{-4}\%$, which is high enough to activate the M1 switch. We then increased the dose of aTc to release the inhibition of C6 production by TetR so that the M2 switch could activate. As shown in Fig. 4b, the M1 switch was activated in the presence of L-ara without aTc. An increase in the dose of aTc did activate the M2 switch as expected, but the M1 switch was then blocked from activation (Fig. 4b and Supplementary Fig. 6a). The simulated cell fates in the space of L-ara and aTc shows that the M1 switch can only be activated with high L-ara and low aTc, while the M2 switch is only activated with high L-ara and high aTc (Supplementary Fig. 6b), which is consistent with the experimental data. These results confirm that the relative strength of the module connections determines the winner of the resource competition in the Syn-CBS circuit.

To prove that the above cell fate transition was from designed fine-tuning of the M1-to-M2 link but not from altered strength of the hybrid promoter Para/tet, we then tested the circuit with hybrid promoter Para/tet but without TetR module (circuit IC25) and observed a similar result as we did with the Syn-CBS circuit CT61 (Supplementary Fig. 7 and Fig. 2). Thus, the change of the promoter sequences did not change the connection strengths between the two modules nor the circuit behavior. We also tested the hybrid Syn-CBS circuit CT81 with a low-copy backbone to study whether the WTA phenomenon could be alleviated. We found that the cell fates were not well separated (Supplementary Fig. 8), which was most likely due to the lower level of circuit's gene products in the activated states and associated higher noise level in the low-copy plasmid system. However, we can still see a similar pattern of resource competition and the WTA phenomenon as we did with the medium-copy plasmid system.

Taken together, although the two bistable switch modules in the Syn-CBS circuit are designed to be mutually activated, they race against each other for the limited resources in order to be activated. That is, the first activated module takes available resources and thus inhibits the activation of the other. Since the

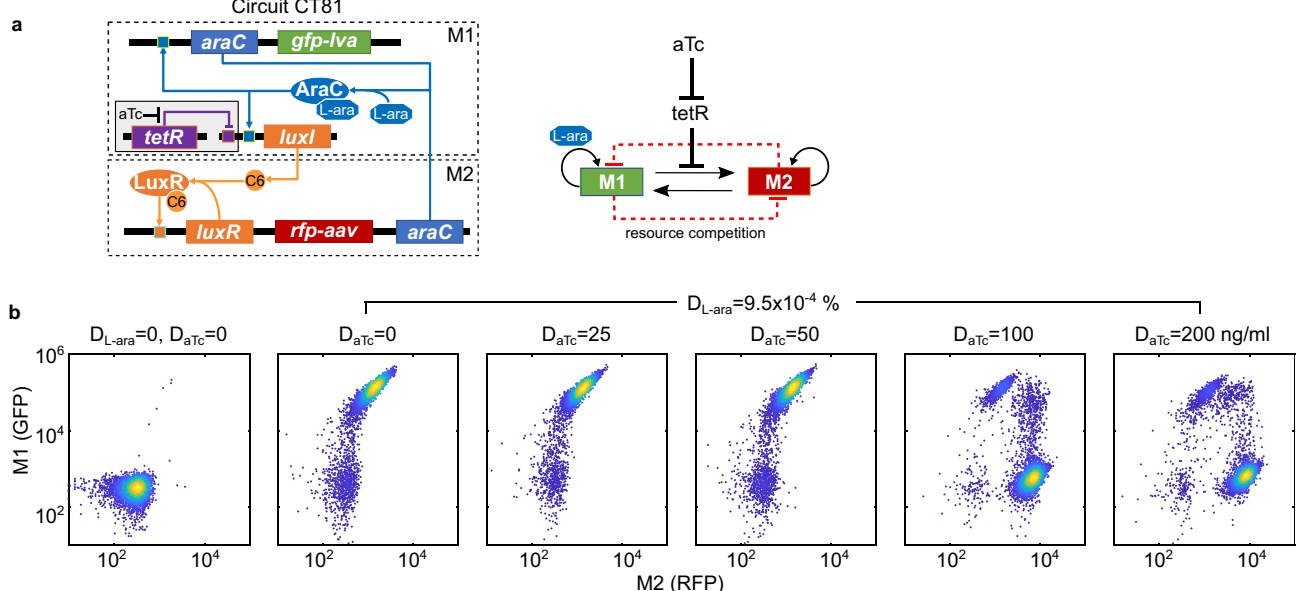

**Fig. 4 Relative strength of module connections determines the winner of resource competition. a** Diagram of the hybrid Syn-CBS circuit with a tetR module for fine-tuning the connection between two bistable switch modules. A hybrid promoter Para/tet is used for controlling the production of C6 to tune the M1-to-M2 connection. **b** Flow cytometry data showed cell state transitions with various doses of inducer aTc ($D_{aTc}$) and a fixed dose of L-ara ($D_{L-ara}$). In total, 10,000 events were recorded for each sample. Data from one representative of five independent biological replicates. Circuit CT81 was used here. Source data are provided as a Source Data file.

Syn-CBS circuit was designed to achieve sequential activation of the two switches, the WTA behavior with the one-strain Syn-CBS system would be a failure in the modularity design of the circuit. Therefore, we need to decouple these indirect hidden links between the modules to achieve sequential activation.

**Stabilized coactivation of the two switches through a division of labor using microbial consortia.** In order to decouple the undesired crosstalk within the gene circuit due to resource competition, we designed and constructed a two-strain Syn-CBS circuit by dividing the two modules into two separate cells (Fig. 5a), instead of placing one whole gene circuit in a single cell. We considered these new two-strain Syn-CBS circuits both with and without the TetR module (Supplementary Fig. 10a and Fig. 5a, respectively) and kept the circuit connections similar to that of the one-strain Syn-CBS circuits (Figs. 1a and 4a). The original M2-to-M1 link cannot be achieved here since the transcriptional factor AraC is not able to travel among cells freely. This link is not required for the functional CBS although it may increase the reversibility of the states as demonstrated by our previous work[16,17].

We systematically studied the cell fate transitions with the two-strain Syn-CBS circuits. For the design without the TetR module (circuit pair CT66 and CT67), a low dose of L-ara was enough to transition some cells into a high-RFP state (Fig. 5b). As the dose of L-ara increased, the rest of the cells gradually transitioned to a high-GFP state and became stable under a high dose of L-ara (Fig. 5b and Supplementary Fig. 9). It is noted that since GFP and RFP are in different strains, stable coactivation of the two modules is instead represented by the coexistence of the RFP[hi]/GFP[lo] and RFP[lo]/GFP[hi] populations (Fig. 5b and Supplementary Fig. 9). Similarly, in the two-strain Syn-CBS circuit with the TetR module (circuit pair CT66 and CT82, Supplementary Fig. 10a), part of the cells first transitioned to a GFP-high state under a low aTc dose. As the aTc dose rises, the rest of the cells continuously transitioned to an RFP-high state (Supplementary Fig. 10b, c). Stable coexistence of the two populations was also found. It is

noted that due to the heterogeneity of the growth rates of the two strains, the fractions of the cell populations were not well-balanced, but the overall ratio did not change over a dose range of the inducer in both circuit pairs. In addition, the weak anticorrelation between the two switches in both two-strain Syn-CBS circuits when under high inducer doses suggests that the adverse effects of resource competition are minimized through a division of labor. Thus, the two-strain Syn-CBS circuits work better to achieve successive activation of the two bistable switches without the result of one being switched off.

## Discussion

Resource competition is commonplace at various levels of regulation in biological systems, including transcriptional, translational, and posttranslational. Resource competition can be exploited to its best advantage for natural and synthetic biological systems. For example, amplified sensitivity arises from covalent modifications with limited enzymes and molecular titration[26–30]. Competition for limited proteases was utilized to coordinate genetic oscillators[31]. Adding competing transcriptional binding sites on sponge plasmids makes the repressilator more robust[32]. However, resource competition within one gene circuit may also change circuit behaviors[7,8]. Here, we showed that it was challenging to achieve successive activation of two bistable switches in one strain due to resource competition. We found that competion for limited resources between the two bistable switches leads to only one winner taking all the available resources. Interestingly, the outcomes of the WTA competition depended on the dynamics of the two switches, given that the faster one was always the winner.

Several approaches have been proposed to counteract the effects of resource competition, either by fine-tuning the parameters in the gene circuit[7,9] or manipulating the size of the orthogonal resource pools[33–36]. Additionally, a burden-driven negative feedback loop was implemented to control gene expression by monitoring the cellular burden[5,37,38]. The negative feedback loop can also be integrated within synthetic gene circuits

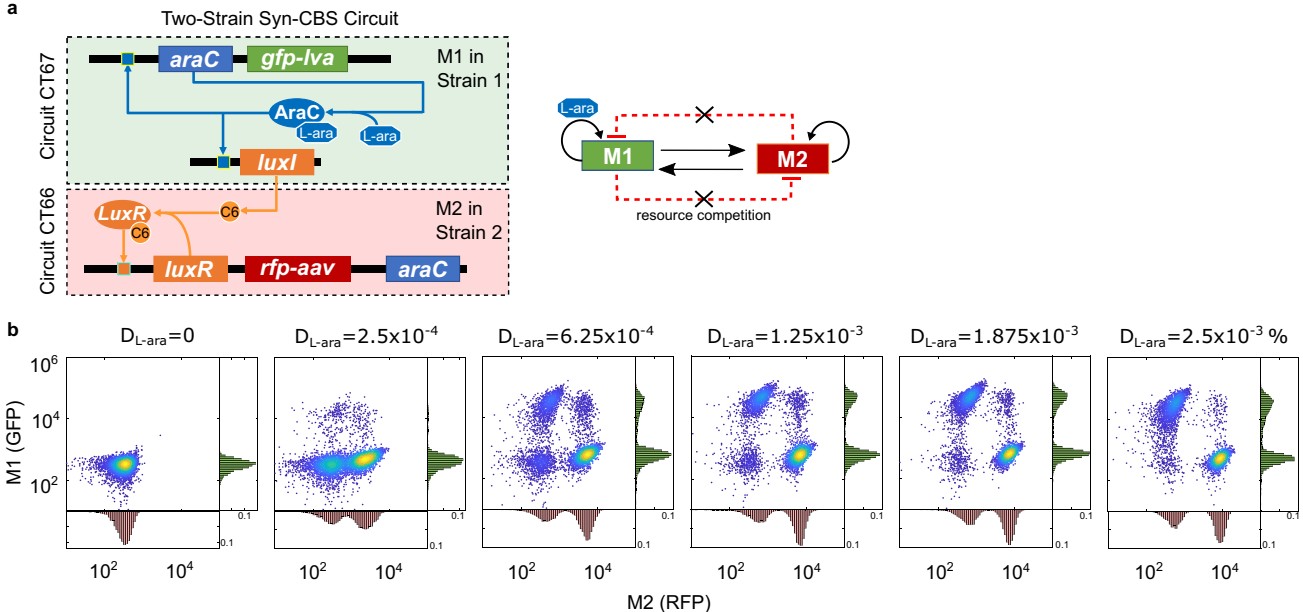

**Fig. 5 Minimize resource competition through a division of labor using microbial consortia. a** Diagram of two-strain Syn-CBS circuits without a tetR module. **b** Flow cytometry data show the expected stepwise cell state transitions by increasing the dose of inducer L-ara ($D_{L-ara}$). In total, 10,000 events were recorded for each sample. Data from one representative of three independent biological replicates. Circuits CT66 and CT67 were used here. Source data are provided as a Source Data file.

to control resource competition[10,39–41]. In this study, we compared the single- and two-strain Syn-CBS circuits and found that the deviated cell fate transitions due to resource competition in monoclonal microbes were corrected in micro-organism consortia. A trade-off was found between robustness to environmental disturbances and robustness to perturbations in available resources for the genetic circuit[42]. Synthetic microbial consortia have been used for engineering multicellular synthetic systems[43–47] and metabolic pathways[48]. Our one-strain Syn-CBS and Syn-SBS circuits can be used to test the other controlling strategies of resource competition. Our two-strain Syn-CBS circuits can be used for studying the multiple cell fate transitions, and potential dynamic yet responsive delivery of multiple drugs.

Resource competition also exists between the host cell and the synthetic gene circuit. Thus, the strategies of the host cells on resource allocation also influence the performance of the gene circuits. Host cells are dynamically adjusting their intracellular resources' reallocations in response to nutrient availability or shift[49–51]. Therefore, the availability of cellular resources to the synthetic gene circuits is also very dynamic and stochastic. Recently, it was found that bacterial strategies differ in their response to starvation for carbon, nitrogen, or phosphate[52,53]. It is very challenging to accurately predict the circuit behaviors under the conditions of dynamic resource allocation. An integrative circuit-host modeling framework has been developed to predict behaviors of synthetic gene circuits[4]. Dynamic models of resource allocation were also developed in response to the presence of a synthetic circuit[54,55].

Our recent work found that synthetic switches may lose memory due to cell growth feedback depending on their network topology[3]. We mathematically and experimentally demonstrated that the self-activation gene circuit is susceptible to the growth feedback. In contrast, the toggle switch circuit is very robust, although the gene expression of both circuits was decreased significantly due to the fast cell growth[3]. Recently, McBride et al. mathematically proved that the mutual activation circuit and reciprocal inhibition circuit also behave differently under the context of resource competition[56]. Similarly, the repression

cascade seems more robust in contrast with the activation cascade[7]. All of these works suggest that the perturbation of the circuit function depends on the network topology, and thus the context of various circuit-host interactions needs to be considered for gene circuit design.

## Methods

**Strains, media, and chemicals**. *E. coli* strain DH10B (Invitrogen, USA) was used for all the cloning and plasmids constructions. *E. coli* strain K-12 MG1655ΔlacI-ΔaraCBAD was used for all the circuits inductions and measurements. The culture media for the cells were LB broth (Luria-Bertani broth, Sigma-Aldrich) or LB plates supplemented with 25 μg/ml chloramphenicol or 50 μg/ml kanamycin depending on the backbone of the plasmids harbored by the cells in question. When plasmid extraction was desired, single DH10B colony carrying the corresponding plasmid was inoculated into 5 ml culture medium and grown in a 15 ml culture tube with 250 revolutions per minute at 37 °C. When circuit induction was performed, MG1655ΔlacIΔaraCBAD carrying the circuit of interest was cultured in 2 ml culture medium supplemented with appropriate inducer in a 15 ml culture tube with 250 revolutions per min at 37 °C. Inducers L-ara (L-(+)-Arabinose, Sigma-Aldrich), C6 (3oxo-C6-HSL, Sigma-Aldrich), and aTc (Anhydrotetracycline hydrochloride, Abcam) were dissolved in ddH$_2$O at concentrations of 25%, 10 mM and 1 mg/ml, and stored at −20 °C in aliquots as stocking solutions. The aTc stocking solutions were replaced every month. When diluted into appropriate working solutions in ddH$_2$O, L-ara, and C6 solutions were replaced monthly, and aTC solutions were prepared freshly each time and discarded after 24 h. All the working solutions were kept at 4 °C and added into culture media with 1000-fold dilution. All the oligo DNAs were synthesized by Integrated DNA Technologies, Inc.

**Plasmids construction**. The araC gene was amplified by PCR using the BioBrick part BBa_C0080 as the template to have the lva tag removed. The primers used were forward 5′-ctggaattcgcggccgcttctagatggctgaagcgcaaaatgatc-3′ and reverse 5′-ggactgcagcggccgctactagtagtttattatgacaacttgacggctacatc-3′. A derivative of Plux named Plux9 was used in this manuscript. The sequence of Plux9 is 5′-acctgtag-gatcgtacagggttacgcaagaaaatggtttgttatatgtcgaataaa-3′. Plux9 was amplified by PCR using the BioBrick part BBa_R0062 as template. The primers used were forward 5′-gcttctagagacctgtaggatcgtacagggttacgcaagaaaatggtttgttatag-3′ and reverse 5′- ggactg-cagcggccgctactagtatttcgactataacaaaccattttc-3′. A derivative of luxR named luxRG2C which harbored two amino acid mutations S116A and M135I was used in this manuscript. Detailed characterization of luxRG2C can be found in[57]. Two sets of primers were used to generate luxRG2C sequence from template BioBrick C0062. Primer set one was forward 5′-ctggaattcgcggccgcttctagatgaaaaaca-taaatgccgac-3′ and reverse 5′-ggactgcagcggccgctactagtagtttattaattttttaaagtatgggcaatc-3′; primer set two was: forward 5′-

gtttagtttccctattcatacggctaacaatggcttcggaatacttagttttgcacattc-3′ and reverse 5′-gtat-gaatagggaaactaaacccagtgataagacctgctgtttttcgcttctttaattac-3′. The gene sequence of unstable RFP tagged with AANDENYAAAV[58] peptide tail (RfpAAV) was synthesized by PCR using BioBrick K1399001 as template and primer set: forward 5′-tgccacctgacgtctaaagaa-3′ and reverse 5′-gctactagtattattaaactgctgctgcgtagttttcgtcgtttgcagc-3′. The sequence of Para/tet is 5′-GCTTCTAGAGacattgattatttgcacggcgtca-cactttgctatgccatagcaagatagtccataagattagcggatcctacctgacgcttttttatcgcaactctctactgtttct ccattccctatcagtgatagaTACTAGTAGCGGCCGCTGCAGTCC-3′, in which the lowercase part stands for the sequence for the promoter and the uppercase part stands for the sequences flank the promoter which can be cut by restriction enzymes XbaI and PstI. All the modified parts were flanked by RFC ten sequence from iGEM in order for them to be constructed the same way as standard Bio-Bricks. The BioBricks used directly to build our circuits were listed in Supplementary Table 1. All parts were first restriction digested using desired combinations of FastDigest restriction enzyme chosen from EcoRI, XbaI, SpeI, and PstI (Thermo Fisher) and separated by gel electrophoresis, and then purified using GelElute Gel Extraction Kit (Sigma-Aldrich) followed by ligation using T4 DNA ligase (New England BioLabs). Then the ligation products were transformed into *E. coli* strain DH10B and later the positive colonies were screened. Finally, the plasmids were extracted using GenElute Plasmids Miniprep Kit (Sigma-Aldrich). Each operon constituting the circuits was constructed monocistronically and its sequence was verified before combined into circuits. Details of all the operons and the circuits can be found in Supplementary Tables 2 and 3. The low-copy assembly backbone pMMB206 was kindly provide by Dr. David Nielsen from Arizona State University. To generate the low-copy assembly of circuits CT61 and CT81, the circuits' fragments were dissected from backbone pSB3K3 with restriction enzymes EcoRI and PstI, and ligated to pMMB206 fragment digested with the same enzyme pair. The gene circuits in this manuscript were all on backbone pSB3K3 unless otherwise stated.

**Flow cytometry**. All samples were analyzed using Accuri C6 flow cytometer (Becton Dickinson) with excitation/emission filters 480 nm/530 nm (FL1-A) for GFP detection and 480 nm/>670 nm (FL3-A) for RFP at indicated time points. In total, 10,000 events were recorded for each sample. At least three replicated tests were performed for each experiment. Data files were analyzed with MATLAB (R2017a, MathWorks). Cells were gated using FSC-A/FSC-H (Supplementary Fig. 11) to eliminate the doublets and noncellular small particles according to data from the plain LB medium without any cells as a negative control.

**Circuit inductions**. The experimental procedure for each biological replicate of the one-strain experiment was carried out like this. On day 1, plasmid carrying the circuit in question was transformed into *E. coli* strain K-12 MG1655ΔlacIΔar-aCBAD which were grown on LB plate with 50 μg/ml kanamycin overnight at 37 °C. On day 2 in the morning, one colony was picked and inoculated into 400 μl LB medium with 50 μg/ml kanamycin and was grown to OD 1.0 (measured in 200 μl volume in 96-well plate by plate reader for absorbance at 600 nm) in a 5 ml culture tube in the shaker. The cells were then diluted 1000 folds into fresh culture medium, and each portion of a 2 ml aliquot was distributed into a 15 ml culture tube. Later, respective inducers were added into each tube, and the cells were grown for 16 h in the shaker till next morning then data were gathered on flow cytometry.

The experimental procedure for each biological replicate of the two-strain experiment was carried out like this. On day 1, each plasmid carrying part of the circuit was transformed into *E. coli* strain K-12 MG1655ΔlacIΔaraCBAD which were grown on LB plate with 50 μg/ml kanamycin overnight at 37 °C. On day 2 in the morning, one colony from each strain was picked and inoculated into 400 μl LB medium with 50 μg/ml kanamycin and was grown to OD 1.0 (measured in 200 μl volume in 96-well plate by plate reader for absorbance at 600 nm) in a 5 ml culture tube in the shaker. Cells from these two strains were then diluted 1000 folds into fresh culture medium in the same tube, and each portion of a 2 ml aliquot was distributed into a 15 ml culture tube. Later, respective inducers were added into each tube, and the cells were grown for 16 h in the shaker till next morning then data were gathered on flow cytometry. Data were analyzed with MATLAB R2017a (MathWorks).

**Average fluorescence analysis performed by plate reader**. Synergy H1 Hybrid Reader from BioTek was used to perform the average fluorescence analysis. In total, 200 μl of culture was loaded into each well of the 96-well plate. LB broth without cells was used as a blank. The plate was incubated at 37 °C with orbital shaking at the frequency of 807 cpm (cycles per minute). Optical density of the culture was measured by absorbance at 600 nm; GFP was detected by excitation/emission at 485/515 nm; RFP was detected by excitation/emission at 546/607 nm.

**Mathematical models**. Ordinary differential equation models were developed to describe and analyze all the synthetic gene circuits with or without consideration of resource competition at the population level. The stochastic simulation algorithm was developed to characterize the stochasticity at the single-cell level. The CME was used to calculate the steady probability distribution and estimate the potential landscape. Details are provided in the Supplementary information.

**Reporting summary**. Further information on research design is available in the Nature Research Reporting Summary linked to this article.

## Data availability
All data produced or analyzed for this study are included in the article and its Supplementary information files or are available from the corresponding authors upon reasonable request. Source data are provided as a Source data file. Plasmids are available through Addgene (ID 165403-165411).

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

## Acknowledgements

We thank David Nielsen and Xuan Wang for providing us the low-copy number plasmid backbones. This project was supported by the ASU School of Biological and Health Systems Engineering and NSF grant (no. EF-1921412 to X-J.T.). H.G. and J.M.-A. were also supported by the Arizona State University Dean's Fellowship.

## Author contributions

R.Z. and X-J.T. conceived the study. X-J.T. and R.Z. designed the study. R.Z. and J. L. performed experiments. J.M.-A., H.G., and X-J.T. performed theoretical studies. J.M.-A., H.G., R.Z., T.D., X.W., and X-J.T. analyzed the data. X-J.T. wrote the manuscript. H.G., R.Z., X.W., and J.M.-A. edited the manuscript.

## Competing interests

The authors declare no competing interests.
