## [Peer Review File · Nature Communications]

Reviewers' Comments:

Reviewer #1:

Remarks to the Author:

Winner-Takes-All Resource Competition Redirects Cascading Cell Fate Transitions by Zhang, et al.

Resource competition in gene expression in cells can be a fundamental bottleneck in achieving desired functionalities of synthetic gene circuits that consist of multiple modules. Authors here uncover a nonlinear resource competition in a synthetic cascading bistable switches circuit (Syn-CBS) with two coupled self-activation modules. They virtually always observed activation of a single module only even when they theoretically predicted a coactivation. They call this phenomenon "winner-takes-all" strategy and found that the winner is determined by the relative strength of the connection between the modules. Furthermore, they overcame this nonlinear resource competition effect by a division of labor – they built a microbial consortium by putting each module in a different strain, and this enabled a stable coactivation of the two modules.

General comments

The work addresses an interesting problem in engineering gene circuits – that is, how the limited host resource can affect gene circuit dynamics and how this effect can be alleviated. The first notion (effects of resource on circuit dynamics) has been well recognized and demonstrated in a few past studies, including the work from their own. The most interesting aspect of the work is the latter point, which is to experimentally demonstrate the use of division of labor to address resource limitation. To my knowledge, this aspect is more novel related to the first point. As I was reading the paper, I wish to see much more in-depth analysis on the latter, instead of spending the major part of the work talking about debugging the circuit in a single strain. In the end, the paper reads like to somewhat disconnected parts. That said, the work is technically strong and its message and presentation can be much strengthened in a revision.

Specific points:

1. It seems that the central objective of the circuit design, in a single strain or in a mixed culture, is to achieve sequential activation of the two bistable modules. It is not entirely clear why this design objective is important, though it can be challenging, as the authors demonstrated. Also, the authors spend way too much text on the debugging the system in the single strain. It's technically strong and the underlying mechanism seems to be quite intuitive.
2. How do they know that what they observed was solely due to resource competition but not other aspects of the circuit design? For example, how would the authors comment on the possible effects of the plasmid copy numbers on their findings? Say they were to use a low copy number plasmid for both of the modules. What would be different? Perhaps, working with a low copy number plasmid could alleviate the "winner-takes-all" phenomenon and makes the co-activation steady-state more accessible?
3. The presentation of modeling analysis needs to be significantly improved. I had a really difficult time understanding the formulation of the ODEs and in particular, how resource competition is implemented, even though I read over the model description multiple times. For example, even the most basic model (the one without resource competition), it is unclear how the two model equations represent positive autoregulation. M1 and M2 don't appear in the definition of f1 or f2. Likewise, it is unclear how resource competition was implemented in the subsequent model.

In the end, I had to take their results at the face value. This is a pity because this was a central point they're making and an intuitive understanding was masked by the complexity (perhaps unnecessary) of the model. In other words, the modeling analysis is not helping a better understanding of the mechanism they're trying to convey.

4. Related to the point above, what is the rationale of the stochastic simulation? What new insights does it provide beyond the simpler phase-plane analysis?

Other points:

Abstract: It sounds like a stable co-activation of the two modules was never observed in single cells of the single strain circuit. However, in the main text, the authors acknowledge that it was observed in a tiny fraction of the population (from the flow cytometry experiments). I believe this discrepancy in presentation may need to be fixed. Moreover, did they attempt to do a time-lapse experiment on the same exact cells to be able to evaluate the stability of the experimentally observed HH state before they concluded that their state is an unstable one?

Page 4 Lines 16-17: It would be nice to explain what those tags were for in GFP-lva and RFP-aav? I believe they were to make them fast-degrading?

Page 5 Line 16: Change "E. Coli" to "E. coli"

Page 5 Lines 18-19: One of the steady-states of their theoretical model is lowGFP and highRFP. So, when the population is dominated by this steady-state we do expect to see an inverse correlation between GFP and RFP signals. I think what authors here did was to measure the signals at increasing arabinose concentrations analogous to the phase plane analysis. But the text does not clearly communicate this.

Fig. 1b-c: - Make the axis labels and legends consistent- choose either M1 or [M1] as the notation to indicate concentration.

- What happens when the relative strength of the connections between the two modules are equal?

Fig. 2e: - Make the axis labels and legends consistent- choose either M1 or [M1] as the notation to indicate concentration.

Page 5 Line 32: Fix the typo in "moduels"

Page 8 Line 2: What do DC6 and TC6 mean? I believe they mean the dose and the time of the addition, respectively, of C6.

Page 9. The way the authors phrase their observations here sounds like every single cell had the possibility of bearing GFP or RFP signals. However, as authors state, there are two cell types one with the GFP circuit and the other one with the RFP circuit.

Page 11 Line 21: Change 'rotations' to 'revolutions'

Page 13 Line 17: Change 'circles' to 'cycles'

Figures 2, 3, 4 and 5: - They lack a clear statement of the number of biological (independent) and technical repeats when applicable.

- I recommend introducing the circuit names (CT61, CT15, CT81, CT66 or CT67) in part (a).

Figures 2 and 3: I recommend being consistent with the chart types for presenting the calculated potential landscapes throughout the figures.

Materials and Methods/ Strains, media, and chemicals: Generally, poorly written. The entire section can benefit from some editing.

Materials and Methods/ Circuit inductions: Generally, poorly written. The entire section can benefit from some editing.

Sup. Information

General comments:

1. What does 'n' mean when number of repeats were reported? Is it technical or biological repeats?
2. What is the reason for using Lara induction for the experiments presented in Ext. Data Fig. 2 and C6 induction for the ones presented in Ext. Data Fig. 3? What would they be seeing if they used exactly the same induction procedure for these two systems?

Extended Data Fig. 1. What is the color code? I believe blue means a higher probability of being in that state while red means a lower probability. Right? I believe a clarification can be helpful.

Extended Data Figs. 2 and 3: What was the concentration of the "other" inducer in these experiments?

Extended Data Fig. 4. (e) The single cell fate given by yellow in the right panel cannot be seen in the phase plane plotted in (d) here. Is this expected? Was what presented in (d) also obtained by stochastic simulations?

Pages 14-15: How did the authors arrive at those parameter choices? How do they justify them? I believe that the phase plane analysis (steady-states and their stability) of the model would critically depend on the parameter choices. So, it is worthwhile to elaborate this further. In Page 14 Line 25, what fitting are they talking about? Did they fit the model to some experimental data and get those parameters? And, how did they know that those are the true parameters they should use?

Page 18 Lines 3-5: But ribosome competition is in downstream of the RNAP competition. So that, once one can achieve a balanced consumption of RNAPs by the two modules, then the ribosome demand will automatically be more balanced too. Right?

Reviewer #2:

Remarks to the Author:

In this study, Zhang et al. started by coupling two bistable switches into one cascading circuit. They expected to create a multistable circuit, in which each switch module activates the other. However, experimentally, only one bistable switch was able to be turned on at a time, while blocking the other, obtaining a 'winner-takes-all' behavior. They hypothesized this behavior was due to resource competition. When they tested a single-strain circuit eliminating interaction among the modules, the resource competition is still present, and will limit expression of both modules together, even though it is still possible to achieve with exogenous control. Similarly, when they split the two modules into separate strains, but keeping the interaction among them, they were able to obtain conditions for activation of both modules. They advocate that a good way to go against resource limitations is to split the work within a microbial consortium to divide the labor. They also tested the single cell circuit further regulated by TetR, which helped control the strength of each module, and predict the winner.

In general, I think this is an interesting study. It provides good insight into general genetic circuit "misbehavior". And even though, microbial consortia worked well, it is not a surprising outcome. In

fact, I thought it was more interesting that you can obtain mutually exclusive states with modules that were supposed to activate each other. I think implications of this finding into Synthetic Biology could be more explored. Or perhaps, new applications for the original circuit.

Furthermore, I had a few questions and points to be addressed that will be listed below:

Major Points:

- 1) As a control, it would be nice to see the circuit with hybrid (ara/tet) promoter tested without TetR present. Because when you change the promoter sequences, the general strength could be different, which could affect circuit behavior. Ideally, without TetR, you would expect to see a similar effect as the non-hybrid circuit.
- 2) To better indicate that the resource competition is the reason for the faulty syn-CBS circuit, you should increase the availability of resources to showcase a functional co-activation of both modules.

Minor Points:

- 3) Why did you use different degradation tags for the reporters? LVA and AAV? What's the reasoning behind it?
- 4) Color selection for nullclines are not great: M2 is the red state, but the red line represents M1. Colors from flow cytometry graphs (fraction of cells) could also be better chosen to help the reader: M1 activation as green, and M2 activation as red, for example. Basically, to match your circuit and facilitate understanding.
- 5) Fig.3C: what's the pink square in the co-activation region? What does it represent?
- 6) In the 2-strain Syn-CBS circuit (Fig. 5), why with an increase in L-arabinose is the first strain to be activated M2? Shouldn't it be M1, because it would require less steps to produce GFP? For RFP to be on, the C6-HSL still needs to be produced. Can it be explained only by the stronger M1-to-M2 link? M1 strain is technically not dependent on M2 anymore, because it produces its own AraC, so there is no M2-to-M1 link?
- 7) Why did you pick the time point of 16h for analysis? How does the OD look at that point? Aren't cells in stationary phase at this point? How might this affect your circuit behavior?
- 8) Why did you decide to put all the components in only one plasmid? Wouldn't it be better to split them into multiple plasmids?
- 9) I noticed some problems in the tables. Some information is missing and possible errors, please revise.

REVIEWER COMMENTS

Reviewer #1 (Remarks to the Author):

Winner-Takes-All Resource Competition Redirects Cascading Cell Fate Transitions by Zhang, et al.

Resource competition in gene expression in cells can be a fundamental bottleneck in achieving desired functionalities of synthetic gene circuits that consist of multiple modules. Authors here uncover a nonlinear resource competition in a synthetic cascading bistable switches circuit (Syn-CBS) with two coupled self-activation modules. They virtually always observed activation of a single module only even when they theoretically predicted a coactivation. They call this phenomenon “winner-takes-all” strategy and found that the winner is determined by the relative strength of the connection between the modules. Furthermore, they overcame this nonlinear resource competition effect by a division of labor – they built a microbial consortium by putting each module in a different strain, and this enabled a stable coactivation of the two modules.

General comments

The work addresses an interesting problem in engineering gene circuits – that is, how the limited host resource can affect gene circuit dynamics and how this effect can be alleviated. The first notion (effects of resource on circuit dynamics) has been well recognized and demonstrated in a few past studies, including the work from their own. The most interesting aspect of the work is the latter point, which is to experimentally demonstrate the use of division of labor to address resource limitation. To my knowledge, this aspect is more novel related to the first point. As I was reading the paper, I wish to see much more in-depth analysis on the latter, instead of spending the major part of the work talking about debugging the circuit in a single strain. In the end, the paper reads like to somewhat disconnected parts. That said, the work is technically strong and its message and presentation can be much strengthened in a revision.

Response: We sincerely thank the Reviewer for appreciating the importance of the resource competition problem in engineering gene circuits, the novelty of the division of labor strategy to address the resource limitation, and the technical strength. We also give thanks for the valuable comments, which have helped us to strengthen the presentation of the key message in the revised manuscript. Below, we have addressed all the comments and questions raised by the Reviewer.

Specific points:

1. It seems that the central objective of the circuit design, in a single strain or in a mixed culture, is to achieve sequential activation of the two bistable modules. It is not entirely clear why this design objective is important, though it can be challenging, as the authors demonstrated. Also, the authors spend way too much text on the debugging the system in the single strain. It's technically strong and the underlying mechanism seems to be quite intuitive.

Response: We thank the Reviewer for this critical comment. One of the motivations of this circuit design is to understand the mechanism of the cascading bistable switches (CBS). In our previous work in Ref. [16, 17], we proposed and then verified that the CBS is the underlying mechanism of epithelial-to-mesenchymal transition (EMT), an essential biological process in embryonic development, tissue regeneration, and cancer metastasis. We mathematically and experimentally demonstrated that there exists a partial EMT state in addition to the epithelial and mesenchymal states in the system and that this two-step process proceeds through stepwise activation of multiple feedback loops [16, 17]. Successive cell fate transition regulated by CBS in this complex biological setting is not easy to be understood. We can use synthetic reconstitution of the CBS circuit using a similar network topology to isolate and define the key players. A synthetic CBS circuit allows us to precisely tune the crucial parameters in the system and study their effects in detail. In addition, the existence of multiple stable states under the same condition, also known as multistability, plays critical roles in diverse biological processes, such as cell differentiation. Thus, our circuit can be used to study the general principle for multiple cell fate transitions.

The reason we spent more text on how resource competition perturbs the system in single strain circuits is that resource competition is one of the most important circuit-host interactions. These circuit-host interactions are regarded as nuisances in the field but are often neglected or oversimplified during the design of genetic devices, leading to a high rate of device failure seen in synthetic biology. Thus, a quantitative understanding of the functional perturbation of gene circuits by resource competition will enable us to control or utilize this interaction to optimize gene circuit functions. There are some studies

on resource competition and its effects on circuit functions, such as Ref. [5-11]. However, these published data show a more linear resource competition. Here in our study, we found a new type of resource competition, which is highly nonlinear and portrays 'winner-takes-all (WTA)' behavior. To us, it was unexpected to obtain mutually exclusive states with only one winner in the one-strain Syn-CBS circuit, in which the two modules were supposed to activate each other. Thus, we spent the major part of the work on this notion, not for debugging the circuit in a single strain, but to characterize this new type of resource competition within one gene circuit.

2. How do they know that what they observed was solely due to resource competition but not other aspects of the circuit design? For example, how would the authors comment on the possible effects of the plasmid copy numbers on their findings? Say they were to use a low copy number plasmid for both of the modules. What would be different? Perhaps, working with a low copy number plasmid could alleviate the "winner-takes-all" phenomenon and makes the co-activation steady-state more accessible?

Response: We thank the Reviewer for this critical comment. Our data indirectly proved the primary contribution to the winner-takes-all dynamic was resource competition by observing circuit behavior from various angles. First, in Fig. 2a, we have shown the negative correlation between the GFP and RFP, which was also found in many other publications on resource competition (Ref. 9-11), even though our data followed a two-phase piecewise linear function. Second, the two modules in the one-strain Syn-CBS circuits (circuits CT61 and CT81) can only be activated exclusively, as shown in Fig. 2 and Fig. 4, respectively. Third, the two separate bistable switches (circuit IC15) could be coactivated when they were activated under similar induction strength, but only one-module activation (one winner) occurs if the strength of one induction is larger than the other (Fig. 3). Fourth, in the system of two separate switches (circuit IC15), we found the 'winner' changed from module 2 to module 1 if the C6 addition was delayed (Supplementary Fig. 5). Fifth, stable coactivation of the two modules in the two-strain Syn-CBS circuit was achieved once each module gained access to more resources. Thus, although we cannot prove that resource competition is the sole factor, we have demonstrated that it is an essential one.

The anticorrelation and the 'winner-takes-all' behavior are not due to other aspects of the circuit design given that no negative regulations were designed between the two modules in the circuit. Following the Reviewer's suggestion, we did additional experiments with a low copy number plasmid for both modules in the Syn-CBS circuits, with and without the TetR module. We found that in the circuit without the TetR module (circuit CT61), Module 1 was not activated even with a high dose of inducer L-ara

(Supplementary Fig. 3). This is most likely due to resource depletion by Module 2. In the circuit with the TetR module (circuit CT81), the cell fates are not well separated (Supplementary Fig. 8), which is most likely due to the low level of molecules in the activated states and the associated high noise level in the low-copy plasmid system. However, we can still see a similar level of resource competition, and the WTA phenomenon as that occurs in the high-copy plasmid system.

Taken together, the resource competition and the WTA phenomenon are not easily resolved by using a low-copy plasmid for the synthetic circuits. In our case, it did not work very well with low-copy plasmid either because one module was not activated properly or the system was too noisy to have well-separated cell fates and clear cell fate transition. That is, this strategy still keeps the circuits from working as designed, especially when medium- or even high-copy plasmid is needed.

In the revised manuscript, we have added Supplementary Fig. 3 and Supplementary Fig. 8, and the descriptions on page 6 line 9-12 and page 9 line 26-32 to demonstrate this point.

Supplementary Fig.3. One-strain Syn-CBS circuit with low-copy backbone is not able to activate Module 1 (M1) due to resource competition. Flow cytometry data shows cell state transitions in a one-strain Syn-CBS circuit with a low-copy backbone by increasing the level of inducer L-ara (D_{L-ara}). 10,000 events were recorded for each sample. Data from one representative of four biological replicates. Circuit CT61 with a low-copy backbone (pMMB206) was used here.

Supplementary Fig.8. One-strain Syn-CBS circuit with low-copy plasmid and tetR module confirmed the resource competition between two modules. Flow cytometry data shows cell state transitions with various doses of inducer aTc (D_{aTc}) and a fixed L-ara dose (D_{L-ara}). 10,000 events were recorded for each sample. Data from one representative of three biological replicates. Circuit CT81 with a low-copy backbone (pMMB206) was used here.

“We also tested this Syn-CBS circuit (circuit CT61) with a low-copy backbone to study whether coactivation could be more accessible. We found that Module 1 was not activated even with a high dose of inducer (Supplementary Fig. 3). This is most likely due to resource depletion by Module 2.”

“We also tested the hybrid Syn-CBS circuit CT81 with a low-copy backbone to study whether the WTA phenomenon could be alleviated. We found that the cell fates were not well separated (Supplementary Fig. 8), which was most likely due to the lower level of circuit’s gene products in the activated states and associated higher noise level in the low-copy plasmid system. However, we can still see a similar pattern of resource competition and the WTA phenomenon as we did with the medium-copy plasmid system.”

3. The presentation of modeling analysis needs to be significantly improved. I had a really difficult time understanding the formulation of the ODEs and in particular, how resource competition is implemented, even though I read over the model description multiple times. For example, even the most basic model (the one without resource competition), it is unclear how the two model equations represent positive autoregulation. M1 and M2 don’t appear in the definition of f1 or f2. Likewise, it is unclear how resource competition was implemented in the subsequent model.

Response: We apologize that the model description was not clear enough. One potential reason could be that we did not model all the genes in the system as separate variables directly. Instead, we modeled the genes under the same promoter as one variable. In this way, we will only need two variables, one variable for each module, and thus are able to do the nullcline and direction field analysis directly. This simplification is reasonable given that the production rates for the genes under the same promoter should be similar since each operon constituting the circuits was constructed monocistronically. It is noted that the GFP and RFP are also the direct reporters of these two variables.

Another potential reason is that some functions depend on M1 and M2 indirectly in the model formulation. We formulated it this way to make the models follow the logic of the above simplification and to follow the same format it does in the general model. If we put everything directly in the functions f1 and f2, it might be even more difficult to understand. Following the Reviewer’s comment, in the revised Supplementary Information, we have added clarification to how the resource competition is implemented and the positive autoregulation are formulated in the ODEs.

In the basic model without resource competition, the autoregulation in two modules is formulated within the functions f1 and f2, in which AraC and LuxR are transcriptional factors for each module and are

dependent on M1 and M2. In the revised Supplementary Information, page 16 lines 1-3 and 5-7, we have added the following sentences.

“It is noted that f_1 is a function of AraC that includes both M1 and M2, thus the positive autoregulation in Module 1 and the connection from Module 2 to Module 1 are formed.”

“Similarly, f_2 is a function of LuxR that includes M2, and thus the positive autoregulation in Module 2 is formed. f_2 is also a function of C6 that includes M1, and thus the connection from Module 1 to Module 2 is formed.”

To represent resource competition within the model, we incorporated a new function into the denominator of the production rate, PF_Q , which includes R1 and R2 (both M1 and M2) thus creating mutual inhibition between the two modules. The format of PF_Q is similar to the classic enzymatic competitive inhibition, including one module as the competitive inhibitor for the other module. We have the derivation of the PF_Q formulation for a general system in the following section. In the revised Supplementary Information, page 17 line 27-31 and page 18 line 1, we have added the following sentences.

“Compared to the above model without resource competition, this model has an additional denominator in the production rate, PF_Q , that is a function of R1 and R2 (see the following section on the general mathematical model for the synthetic circuit with resource competition). In the functions of R1 and R2, the levels of transcriptional factors AraC and LuxR depend on M1 and M2, respectively, thus creating mutual inhibition between the two modules as a result of resource competition.”

In the end, I had to take their results at the face value. This is a pity because this was a central point they're making and an intuitive understanding was masked by the complexity (perhaps unnecessary) of the model. In other words, the modeling analysis is not helping a better understanding of the mechanism they're trying to convey.

Response: We understand the mathematical models have some level of complexity, which is necessary in our opinion although it may make the key point of this work a bit nonintuitive. However, we are trying to provide a mechanistic understanding of the 'winner-takes-all' (WTA) behavior. We agree with the Reviewer that it is easy to understand the phenomena from the experimental data directly, but we will lack a mechanistic explanation if there is no supportive modeling component. In addition, the published data in Ref. [5-11] showed more linear resource competition. If we follow the logic of this linear resource competition theory, we are not able to understand the WTA phenomena found here. The analysis from

the model without resource competition shows three stable steady states for the systems and how it proceeds from no activation to one module activation and then to coactivation. The analysis from the model with resource competition explains why the system has different stable steady states without the coactivation state and how the two modules exclude each other from activation. Thus, the modelling analysis provides a mechanistic understanding, and we wish to keep the modeling analysis in the manuscript even if it adds some level of complexity.

4. Related to the point above, what is the rationale of the stochastic simulation? What new insights does it provide beyond the simpler phase-plane analysis?

Response: We used stochastic simulation to demonstrate the stochastic cell fate transitions. For example, deterministic simulation in Fig. 3c shows only one stable steady state under one experimental condition. While the stochastic simulation is able to show that potential transition to other state steady states due to the stochasticity in the system. This is consistent with the flow cytometry data, where multiple states can often be observed under a single experimental condition.

The nullclines and the directed field analysis allows us to see all the stable steady states. However, under one specific initial condition, we are not always able to observe all of them experimentally. To show this, we have updated Fig. 3d and Supplementary Fig. 5. The revised Fig. 3d now shows the nullcline and directed field analyses for the condition with the fixed L-ara does and high C6 does, where three steady states (green, yellow, and red circles) can be found. However, there is little chance to have the cell in the low-RFP/high-GFP state given that the system is set initially at the steady state without any inducers (purple circle), which is closer to the high-RFP/low-GFP state. Thus, two major stochastic trajectories were observed, as shown in Fig. 3d. This is consistent with the flow cytometry data of Fig. 3b (rightmost panel). In Supplementary Fig. 5b, however, we see that the experimental procedure is more likely to set the system at the differential initial state when L-ara is added first. Thus, the system can be triggered into the low-RFP/high-GFP state, which was also represented by the three representative stochastic trajectories in Supplementary Fig. 5e.

In the revised manuscript, we have changed the following sections on page 7 line 20-27.

“Three stable steady-states, including two states with only one winner and the coactivation state can be found at the intersections of the nullclines and direction field (green, red and yellow circles, Fig. 3d) and in the dark blue areas of the potential landscape (Fig. 3e) using inducer doses at $9.5 \times 10^{-4}\%$ for L-ara and $5 \times 10^{-8} M$ for C6. However, only two single-cell representative trajectories simulated from the stochastic model (red and yellow highlighted trajectories Fig. 3d, see SI for more details) are found to be

in either the coactivation or M2 activation state. This is consistent with the flow cytometry data (Fig. 3b, rightmost panel)."

Other points

Abstract: It sounds like a stable co-activation of the two modules was never observed in single cells of the single strain circuit. However, in the main text, the authors acknowledge that it was observed in a tiny fraction of the population (from the flow cytometry experiments). I believe this discrepancy in presentation may need to be fixed. Moreover, did they attempt to do a time-lapse experiment on the same exact cells to be able to evaluate the stability of the experimentally observed HH state before they concluded that their state is an unstable one?

Response: We thank the Reviewer for this helpful comment. In the revised manuscript, we fixed the presentation on the interpretation of the flow cytometry data since. We wished to say that this high-RFP/high-GFP state is not a steady state. We changed the sentence to "*experimental data showed a negligible chance for the existence of the theoretically expected coactivation state (high-RFP/high-GFP), which we no longer consider a steady state.*" on page 6 line 6-8. This negligible number of cells could be from the doublets of small green/red cells, from the rare large noise in some cells, or from debris particles. Given that the fraction is super small compared with other stable steady states across all inducer dose concentrations, it is very reasonable to not consider it as a steady state in the system. Unfortunately, we are not able to evaluate the stability of this state in experiment at current stage due to two reasons. First, there are only a few cells in this quadrant, so the chances that we could trace and observe the coactivated cells under the microscope for a time-lapse experiment would be extremely low. Second, the growth condition would be different from the current experimental setting if we did a time-lapse experiment in the microfluidics, which may introduce altered dynamics to the cells.

Page 4 Lines 16-17: It would be nice to explain what those tags were for in GFP-lva and RFP-aav? I believe they were to make them fast-degrading?

Response: Yes, these two tags are used to make the GFP and RFP fast degrading. We chose these for our circuit because we wanted to make sure the maintenance of GFP/RFP in the stable steady state is not from their slow degradation. We have a short clarification in the revised manuscript on page 4, line 18-19.

"Both tags were chosen to ensure that maintenance of stable steady states was not due to GFP or RFP slow degradation."

Page 5 Line 16: Change "E. Coli" to "E. coli"

Response: This typo has been corrected. A same typo in the Method section was also corrected.

Page 5 Lines 18-19: One of the steady-states of their theoretical model is lowGFP and highRFP. So, when the population is dominated by this steady-state we do expect to see an inverse correlation between GFP and RFP signals. I think what authors here did was to measure the signals at increasing arabinose concentrations analogous to the phase plane analysis. But the text does not clearly communicate this.

Response: We thank the Reviewer for this helpful comment. We agree with the Reviewer that in Fig. 2a we measured the signals at increasing arabinose concentrations analogous to the phase plane analysis. With increase of the arabinose concentration, theoretically the output level of one module should increase and in turn its activation will further increase the output of the other module Therefore, we expected a positive correlation between the gene expression in the two modules. When arabinose is low, we see RFP dominating, which should correspond to the $RFP^{high}GFP^{low}$ state. As the arabinose increases, we expected to see more and more cells in $RFP^{high}GFP^{high}$ and a positive correlation between RFP and GFP. However, experimentally we saw that an increase in GFP corresponds to a decrease in RFP, creating a negative correlation (Fig. 2b). In the revised manuscript, we have added more interpretation for this figure on page 5 and line 24-30.

"We first studied the relationship between the two modules by measuring the mean GFP and RFP levels at increasing arabinose concentrations analogous to the phase plane analysis using a plate reader. We found that RFP vs. GFP showed a negative relationship, as increase of one module simultaneously decreases the other (Fig. 2a). These results are opposite from the theoretical analysis that both modules positively activate each other. This suggests that there is significant resource competition between the two modules."

Fig. 1b-c: - Make the axis labels and legends consistent- choose either M1 or [M1] as the notation to indicate concentration.

Response: This typo has been corrected by changing legend [M1] to M1.

- What happens when the relative strength of the connections between the two modules are equal?

Response: If we have exact equal strength of the two connections between two modules, the activation thresholds for two modules will be the same. We could have two possibilities. First, the activation

thresholds are small so the system has two stable steady states from the intersection of the nullclines, a RFP^{low}/GFP^{low} state and a RFP^{high}/GFP^{high} state. The second possibility is that the activation thresholds are large and the system has four stable steady states from the intersection of the nullclines: a RFP^{low}/GFP^{low} state, a RFP^{high}/GFP^{low} state, a RFP^{low}/GFP^{high} state, and a RFP^{high}/GFP^{high} state. However, these two cases are not good designs for achieving successive cell fate transitions. Thus, we did not consider these two cases to show the design principle of the Syn-CBS circuit. We have added a short discussion on this in the revised manuscript on page 5 line 16-19.

“When the relative strength of the connections between the two modules is similar, we could have either a system with only the two LL and HH states, or a system with all the four states, both of which are not good for successive cell fate transitions.”

Fig. 2e: - Make the axis labels and legends consistent- choose either M1 or [M1] as the notation to indicate concentration.

Response: This typo has been corrected by changing legend [M1] to M1. Same revision was done for Fig. 3 and Supplementary Fig. 5.

Page 5 Line 32: Fix the typo in “moduels

Response: This typo has been corrected.

Page 8 Line 2: What do DC6 and TC6 mean? I believe they mean the dose and the time of the addition, respectively, of C6

Response: Yes. D_{C6} and T_{C6} mean the dose and the time of the addition of C6, respectively. Following Reviewer’s suggestion, we made this clear in the figure caption and the main text in the revised manuscript.

Page 9. The way the authors phrase their observations here sounds like every single cell had the possibility of bearing GFP or RFP signals. However, as authors state, there are two cell types one with the GFP circuit and the other one with the RFP circuit.

Response: We rephrased this description on page 10 line 24-27,

‘It is noted that since GFP and RFP are in different strains, stable coactivation of the two modules is instead represented by the coexistence of the RFP^{hi}/GFP^{lo} and RFP^{lo}/GFP^{hi} populations (Fig. 5b, Supplementary Fig. 9).’

Page 11 Line 21: Change 'rotations' to 'revolutions

Response: This typo has been corrected.

Page 13 Line 17: Change 'circles' to 'cycles

Response: This typo has been corrected.

Figures 2, 3, 4 and 5: - They lack a clear statement of the number of biological (independent) and technical repeats when applicable

Response: We have added the statement of the number of biological repeats in Fig. 2-5.

- I recommend introducing the circuit names (CT61, CT15, CT81, CT66 or CT67) in part (a).

Response: We thank Reviewer for the great suggestion. We have added the circuit names in part (a) of each figure.

Figures 2 and 3: I recommend being consistent with the chart types for presenting the calculated potential landscapes throughout the figures.

Response: We have revised the chart type of the potential landscape in Fig. 2.

Materials and Methods/ Strains, media, and chemicals: Generally, poorly written. The entire section can benefit from some editing

Response: We have fully revised this part.

Materials and Methods/ Circuit inductions: Generally, poorly written. The entire section can benefit from some editing.

Response: We have fully revised this part.

Sup. Information

General comments:

1. What does 'n' mean when number of repeats were reported? Is it technical or biological repeats?

Response: 'n' is the biological repeats in all of our experiments. We updated this in all the figure captions.

2. What is the reason for using Lara induction for the experiments presented in Ext. Data Fig. 2 and C6 induction for the ones presented in Ext. Data Fig. 3? What would they be seeing if they used exactly the same induction procedure for these two systems?

Response: Thanks for pointing this out. Two different circuits were used in Ext. Data Fig. 2 (now as Supplementary Fig. 2) and Ext. Data Fig. 3 (now as Supplementary Fig. 4). The cascading bistable switches (Syn-CBS) Circuit (circuit CT61) was used in Supplementary Fig. 2, which is an extension of Fig. 2 and shows the transition from no activation to M2 activation and then to M1 activation. There is only one inducer, L-ara, needed for the Syn-CBS Circuit (circuit CT61) to activate both modules, since the two modules are connected. Specifically, the inducer for M1 activation is L-ara and the inducer for M2 activation is C6. Since M1 can produce C6 for M2, we only need L-ara to activate the whole circuit.

The two separate bistable switches circuit (circuit IC15) was used in Supplementary Fig. 4, which is an extension of Fig. 3. In the circuit IC15, the two modules are disconnected, so we need both inducer L-ara and C6 to activate the whole circuit. The purpose of Fig. 3 is to show the transition from no activation to M1 activation and then to M2 activation, so we decided to increase the C6 dose and kept the L-ara dose fixed. We have clarified this in the revised Supplementary Information and have marked the L-ara dose in Supplementary Fig. 4.

If we induced both the two systems the way as we did in Supplementary Fig. 2, we would not be able to see the M2 activation but only the M1 activation in the two separate bistable switches circuit (circuit IC15) because there is no C6 production or addition. Meanwhile, if we induced both systems the way as we did in Supplementary Fig. 4, we would probably see M2 as the only winner in the syn-CBS circuit (circuit CT61) because we added extra C6 to the system in addition to the levels produced by the M1 module.

Extended Data Fig. 1. What is the color code? I believe blue means a higher probability of being in that state while red means a lower probability. Right? I believe a clarification can be helpful.

Response: That is right. Blue means a higher probability of being in that state while red means a lower probability. Following the Reviewer's suggestion, we made this clear in the figure caption in the revised manuscript. As well, anytime the figure is mentioned outside of the figure caption, we added clarification that the dark blue corresponds to the steady states (page 5 line 13 and page 6 line 25).

Extended Data Figs. 2 and 3: What was the concentration of the "other" inducer in these experiments?

Response: In Supplementary Fig. 2, there is only one inducer, which is L-ara and its dose is shown in the x-axis ticks. In Extended Data Fig. 3 (now as Supplementary Fig. 4), the L-ara dose is fixed as 9.5×10^{-7}

⁴% and C6 dose is shown in the x-axis ticks. We have revised the Supplementary Fig. 4 and its caption to include this information.

Extended Data Fig. 4. (e) The single cell fate given by yellow in the right panel cannot be seen in the phase plane plotted in (d) here. Is this expected? Was what presented in (d) also obtained by stochastic simulations?

Response: Thanks to the Reviewer for this comment. Extended Data Fig. 4 is now Supplementary Fig. 5. The data presented in Supplementary Fig. 5d is from the simulation with the ODE model. Supplementary Fig. 5e shows the potential single cell trajectories from stochastic simulations. To make these two figures consistent, we have updated Fig.4 and Supplementary Fig. 5. Now Supplementary Fig. 5 captures the yellow trajectory which shows that a short delay for the induction of C6 leads to coactivation, which is consistent with Supplementary Fig. 5d. It is noted that due to the stochasticity in the system, the system has the potential to be triggered to all of the three states even if only one state can be found using a combination of C6 dose and induction time with the ODE simulation (Supplementary Fig. 5d).

Supplementary Fig.5. Resource competition between two separate bistable switches with sequential addition of the two inducers. d Simulated cell fates in the space of the dose and timing of inducer C6. L-ara dose was fixed as $D_{L-ara}=9.5 \times 10^{-4}$ %. **e** Simulated stochastic trajectories in the phase plane diagram. The initial state of the cells is set to the steady-state without any inducer (purple circle). The nullclines of M1 and M2 are shown in green and red, respectively, while separatrices are shown in pink. The three cell fates (red, green, and yellow circles) are found at the intersections of the two nullclines. The vector field of the system is represented by small arrows, where the color is proportional to the field strength. Three representative single-cell stochastic trajectories (highlighted green, yellow, and red) show the evolution of the system from the same initial condition ($D_{L-ara}=0\%$ and $D_{C6}=0$ M) to three different states with various T_{C6} . The dose of C6 was fixed as $D_{C6}=0$ M in the left panel and $D_{C6}=5 \times 10^{-8}$ M in the right panel. L-ara was fixed as $D_{L-ara}=9.5 \times 10^{-4}$ % in both panels. Circuit IC15 was used here.

Pages 14-15: How did the authors arrive at those parameter choices? How do they justify them? I believe that the phase plane analysis (steady-states and their stability) of the model would critically depend on the parameter choices. So, it is worthwhile to elaborate this further. In Page 14 Line 25, what fitting are they talking about? Did they fit the model to some experimental data and get those parameters? And, how did they know that those are the true parameters they should use?

Response: Thanks to the Reviewer for this comment. Most of the parameters are based on our model for the AraC module (M1) in our previous work in Ref. [3], where we fitted the parameters with the experimental data, such as the dose-response curve of the promoter. The parameters for the LuxR modules (M2) are set similar to the AraC module. For some parameters, such as the connection strength, we were not able to measure them experimentally. Thus, we considered all the potential possibilities based on our design. For example, we studied two theoretical possibilities to achieve successive cell fate transitions, either from LL (low-RFP/low-GFP), to LH (low-RFP/high-GFP), then to HH (high-RFP/high-GFP), or from LL (low-RFP/low-GFP), to HL (high-RFP/ low-GFP), then to HH (high-RFP/high-GFP). We also experimentally found these two possibilities in Fig. 2 and Fig. 4, respectively. For the two separate switches system, we fitted the experimental data in Fig. 3 and Supplementary Fig. 3, including the threshold of the C6 dose and addition time for the cell fate transition. Unfortunately, we are not able to measure each parameter individually under the current lab condition, and thus we do not know if these are the true parameters.

Page 18 Lines 3-5: But ribosome competition is in downstream of the RNAP competition. So that, once one can achieve a balanced consumption of RNAPs by the two modules, then the ribosome demand will automatically be more balanced too. Right?

Response: Yes. The ribosome competition is in the downstream of RNA competition. We agree with the Reviewer that if the RNAP is balanced to two modules, then the ribosome demand will automatically be more balance as well. This is true if the two modules are activated at almost the same time. However, in the design of our one-strain Syn-CBS circuit, it is not easy to have both RNAP and ribosome balanced in both modules given that one module depends on the activation of the other. That is why we highlighted it in the manuscript that the sequential or time-dependent module activation is challenging due to the resource competition.

Reviewer #2 (Remarks to the Author):

In this study, Zhang et al. started by coupling two bistable switches into one cascading circuit. They expected to create a multistable circuit, in which each switch module activates the other. However, experimentally, only one bistable switch was able to be turned on at a time, while blocking the other, obtaining a 'winner-takes-all' behavior. They hypothesized this behavior was due to resource competition. When they tested a single-strain circuit eliminating interaction among the modules, the resource competition is still present, and will limit expression of both modules together, even though it is still possible to achieve with exogenous control. Similarly, when they split the two modules into separate strains, but keeping the interaction among them, they were able to obtain conditions for activation of both modules. They advocate that a good way to go against resource limitations is to split the work within a microbial consortium to divide the labor. They also tested the single cell circuit further regulated by TetR, which helped control the strength of each module, and predict the winner.

In general, I think this is an interesting study. It provides good insight into general genetic circuit "misbehavior". And even though, microbial consortia worked well, it is not a surprising outcome. In fact, I thought it was more interesting that you can obtain mutually exclusive states with modules that were supposed to activate each other. I think implications of this finding into Synthetic Biology could be more explored. Or perhaps, new applications for the original circuit.

Response: We sincerely thank the Reviewer for appreciating the importance of the study, the insight gained here into general genetic circuit 'misbehavior' and the of interesting 'winner-takes-all' phenomena. Following Reviewer's valuable suggestions and comments, we have added some discussion on the implication of our finding into synthetic biology and potential application of our circuit on page 11 line 29-32.

"Our one-strain Syn-CBS and Syn-SBS circuits can be used to test the other controlling strategies of resource competition. Our two-strain Syn-CBS circuits can be used for studying the multiple cell fate transition, and potential dynamic yet responsive delivery of multiple drugs."

Below please find our response to all the issues raised by the Reviewer.

Furthermore, I had a few questions and points to be addressed that will be listed below:

Major Points:

1) As a control, it would be nice to see the circuit with hybrid (ara/tet) promoter tested without TetR present. Because when you change the promoter sequences, the general strength could be different, which could affect circuit behavior. Ideally, without TetR, you would expect to see a similar effect as the non-hybrid circuit.

Response: Thanks to the Reviewer for this comment. Following Reviewer's suggestion, we built and tested the circuit (circuit IC25) with hybrid (ara/tet) promoter but without TetR present and observed a similar effect as to that of the non-hybrid circuit CT61 (Supplementary Fig. 7, Fig. 2). We see clear resource competition between the two modules and WTA behavior. Thus, the change of the promoter sequences does not change the connection strengths between the two modules nor the circuit behavior.

In the revised manuscript, we added Supplementary Fig. 7 and description on page 9 line 22-26 to demonstrate this point.

Supplementary Fig.7. One-strain Syn-CBS circuit with hybrid promoter but without TetR module confirmed the resource competition between two modules. Flow cytometry data shows cell state transitions with various doses of inducer L-ara (DL-ara). 10,000 events were recorded for each sample. Data from one representative of four biological replicates. Circuit IC25 was used here.

“... we then tested the circuit with hybrid promoter Para/tet but without TetR module (circuit IC25) and observed a similar result as we did with the Syn-CBS circuit CT61 (Supplementary Fig. 7, Fig. 2). Thus, the change of the promoter sequences did not change the connection strengths between the two modules nor the circuit behavior.”

2) To better indicate that the resource competition is the reason for the faulty syn-CBS circuit, you should increase the availability of resources to showcase a functional co-activation of both modules.

Response: Thank the Reviewer for this comment. It will be ideal for us to increase the resource to show that the one-strain Syn-CBS circuit works with coactivation of both modules. However, in our current experimental conditions, we are not able to increase the cellular resource availability. Our data already indirectly proved this through various angles. First, in Fig. 2a, we have shown the negative correlation between GFP and RFP, which was also found in many other publications on resource competition (Ref. 9-11), even though here our data followed a two-phase piecewise linear function. Second, the two modules in the one-strain Syn-CBS circuits (circuits CT61 and CT81) can only be activated exclusively as shown in Fig. 2 and Fig. 4, respectively. Third, two separate modules (circuit IC15) can be coactivated when they are activated under similar induction strength but there is only one module activation (one winner) if the induction strength of one module is larger than the other one (Fig. 3). Fourth, in the system of two separate switches (circuit IC15) with delayed addition of inducer C6, we found the ‘winner’ changed from Module 2 to Module 1 (Supplementary Fig. 5). Fifth, stable coactivation of the two modules in the two-strain Syn-CBS circuit was achieved once each module gained access to more resources. Thus, although we are not able to prove that resource competition is the sole factor, we can say it is a very important one.

Minor Points:

3) Why did you use different degradation tags for the reporters? LVA and AAV? What’s the reasoning behind it?

Response: These two tags are used to make the GFP and RFP fast degrading. We choose these in our circuit because we want to make sure the maintenance of GFP/RFP in the stable steady state is not from their slow degradation. Proteins with LVA tag are degraded a bit faster than the ones are degraded with AAV tag. We initially used LVA tag for both GFP and RFP, but we later discovered that RFP with LVA tag were very faint when detected with our flow cytometer. We then switched to AAV tag for RFP to make it more stable so that we could detect it distinctively. We have added a short clarification in the revised manuscript on page 4, line 18-19.

“Both tags were chosen to ensure that maintenance of stable steady states was not due to GFP or RFP slow degradation.”

4) Color selection for nullclines are not great: M2 is the red state, but the red line represents M1. Colors from flow cytometry graphs (fraction of cells) could also be better chosen to help the reader:

M1 activation as green, and M2 activation as red, for example. Basically, to match your circuit and facilitate understanding.

Response: Thanks for the suggestion for the color selection. In the revised manuscript, we revised the line color of the nullclines in Fig. 1-3 and Supplementary Fig. 5, and the bar color of the cell fractions in Supplementary Fig. 2, 4, 5, 6, 9, 10.

5) Fig.3C: what's the pink square in the co-activation region? What does it represent?

Response: It was a typo. It has been removed in the revised figure 3.

6) In the 2-strain Syn-CBS circuit (Fig. 5), why with an increase in L-arabinose is the first strain to be activated M2? Shouldn't it be M1, because it would require less steps to produce GFP? For RFP to be on, the C6-HSL still needs to be produced. Can it be explained only by the stronger M1-to-M2 link? M1 strain is technically not dependent on M2 anymore, because it produces its own AraC, so there is no M2-to-M1 link?

Response: Thanks for pointing this out. Yes, from our data in both Fig. 5 and Fig. 2, the link from M1-to-M2 link is strong so that M2 can be more easily activated than M1. There is no M2-to-M1 link for the two-strain Syn-CBS circuit as the original link was not designed by the quorum-sensing signal. In addition, this link is not required for the functional cascading bistable switches although it may increase the reversibility of the systems as demonstrated by our previous work in Ref. [16]. In the revised manuscript, we have added some discussion on this point on page 10 line 15-19.

"The original M2-to-M1 link cannot be achieved here since the transcriptional factor AraC is not able to travel among cells freely. This link is not required for the functional cascading bistable switches although it may increase the reversibility of the states as demonstrated by our previous work [16]."

7) Why did you pick the time point of 16h for analysis? How does the OD look at that point? Aren't cells in stationary phase at this point? How might this affect your circuit behavior?

Response: Yes, we agree with the Reviewer that at the time point of 16h, the host cells are already in the stationary phase. At this time point, the system (including the cell growth) reaches the steady states, which is good for us to study the cell fate transitions directed by our circuits. In the previous work of ours (Ref. 3) and others (Ref. 59), there is a long period of constant protein production activity during the stationary phase of bacteria. Thus, our system being at this early stage of stationary phase is good for us to study resource competition. In addition, fast cell growth may induce the memory loss of the bistable

switch, especially the self-activation modules in our circuit, as we found previously (Ref. 3). The growth feedback is out of the scope of this study. All the experiments are compared at this same time point, so that resource limitation is the same for all of the circuits. Furthermore, we have well-designed control experiments in each figure. Thus, our conclusion is solid under the current experimental setting.

8) Why did you decide to put all the components in only one plasmid? Wouldn't it be better to split them into multiple plasmids?

Response: Splitting the circuit into multiple plasmids may cause problems such as loss of the ratio balance of two modules or even loss of one module as the cells divide, which will cause significant noise and uncertainty into the system. Thus, we decide to put all the components into one plasmid.

9) I noticed some problems in the tables. Some information is missing and possible errors, please revise.

Response: In the revised manuscript, we updated Supplementary Table 1-4.

Reviewers' Comments:

Reviewer #1:

Remarks to the Author:

In the revised manuscript, the authors have provided additional experimental data to further corroborate their major conclusion, which is that resource competition is most likely underlying their observation of the "winner takes all" dynamics in a two-module gene circuit. They have also provided further clarifications on the rationale and formulation of the corresponding mathematical models. As a result, the manuscript has been substantially improved and I support its publication in the journal.

Reviewer #2:

Remarks to the Author:

I am satisfied with the revised manuscript and am supportive of its publication.

REVIEWERS' COMMENTS

Reviewer #1 (Remarks to the Author):

In the revised manuscript, the authors have provided additional experimental data to further corroborate their major conclusion, which is that resource competition is most likely underlying their observation of the "winner takes all" dynamics in a two-module gene circuit. They have also provided further clarifications on the rationale and formulation of the corresponding mathematical models. As a result, the manuscript has been substantially improved and I support its publication in the journal.

Response: Thanks to the Reviewer for taking the time to read the revised manuscript carefully and finding the revised manuscript improved substantially. We appreciate the Reviewer's support of its publication in Nature Communications.

Reviewer #2 (Remarks to the Author):

I am satisfied with the revised manuscript and am supportive of its publication.

Response: Thanks to the Reviewer for taking the time to read the revised manuscript carefully. We are glad to hear the Reviewer is satisfied with the revised manuscript and supportive of its publication in Nature Communications.